# AUTOTUNE FOR TIME SERIES TRANSFORMERS USING LOW RANK ADAPTATION AND LIMITED DISCREPANCY SEARCH

## ABSTRACT

Transformer models have achieved remarkable results in the field of Natural Language Processing (NLP) with the introduction of breakthrough large language models like GPT and LLaMA recently. Motivated by their ability to capture long-range dependencies, researchers have successfully adapted these models to the task of time series forecasting. However, despite their potential, effectiveness of applying these pre-trained time series transformer models in the target domain is limited due to the need for hyper-parameter optimisation to match the characteristics of the target domain. This paper presents a novel algorithm that uses parameter efficient fine-tuning such as Low Rank Adaptation (LoRA) coupled with Limited Discrepancy Search (LDS) to efficiently auto fine-tune pre-trained time series transformers for a given target domain. Our approach helps in making informed design choices involving LoRA tunable hyper-parameters with strong performance-cost trade-offs that are highly transferable across different target domains. Our experiments demonstrate that autotune efficiently identifies the optimal configuration of LoRA hyper-parameters, achieving an average MASE improvement of 5.21% across all datasets and 4.76% for out-of-domain datasets compared to zero shot pre-trained models, with improvements as high as 20.59% for one of the out-of-domain datasets.

## 1 INTRODUCTION

Time series forecasting has always been critical for decision-making across various domains including retail, smart grids, healthcare, finance, weather, traffic control among others Peterson (2017) Hernandez et al. (2014). Traditionally, the task of time series forecasting was accomplished using classical statistical models like ARIMA followed by more modern approaches like machine learning (ML) models including Recurrent Neural Network (RNN) and Long Short-Term Memory (LSTM) models Hochreiter & Schmidhuber (1997). However, the remarkable success of LLMs in broad domains like Computer Vision (CV) and Natural Language Processing (NLP) Vaswani et al. (2017) Wen et al. (2022a) has prompted researchers to adapt these models to the task of time series forecasting Wu et al. (2022); Garza & Mergenthaler-Canseco (2023); Gruver et al. (2024b) given their ability to capture long range dependencies present in time series data. These models are pre-trained on vast amounts of data spanning a multitude of domains leveraging the general purpose representations learnt in the process. However, to excel on the domain specific downstream task it becomes important to fine-tune these models on datasets from the target domain Wen et al. (2022b). Such adaptation is generally achieved via fine-tuning, which involves updating all the parameters of the pre-trained model Jin et al. (2023); Bommasani et al. (2021); Lv et al. (2023). Given the large number of parameters that are originally trained, full fine-tuning becomes an operational challenge if we wish to adapt these models to the target domain.

Recent studies have shown that this issue can be mitigated by adapting only a small percentage of the parameters in addition to the pre-trained model for each task and greatly enhance the operational efficiency of these models Hu et al. (2021b). Then, the process of fine-tuning involves taking a pre-trained model and updating only a subset of the model weights to learn the specific characteristics of the target-domain. These techniques are termed as Parameter-Efficient Fine-Tuning (PEFT). One of the popular PEFT techniques, LoRA (Low Rank Adaptation)Hu et al. (2021b) trains only selective

dense layers in a neural network while keeping the pre-trained weights frozen. This approach has been quite widely applied to different domains like medical imaging, video text generation and speech synthesis Balne et al. (2024) to name a few. However, its application to time series is still in its nascent stage.

Further, AutoML which involves automating the process of composing and parameterizing ML algorithms to maximize a specific metric such as model accuracy on a given dataset has been widely used to improve the accuracy of traditional machine learning and deep learning models He et al. (2021). Leveraging this knowledge, autotuning of pre-trained transformers for a given target domain can potentially lead to improvement in the accuracy of transformers compared to traditional fine-tuning with fixed hyper-parameters of the fine-tuning algorithms. However, despite its popularity and widespread use, optimizing an ML pipeline poses significant challenges like slow training and large computational overheads as the search space increases Tornede et al. (2023).

Therefore, with this work, we envision efficient autotuning pre-trained transformers through the integration of LoRA and AutoML with an aim to improve the performance of time series transformers in the target domain. In particular, to achieve an efficient implementation of autotuning, we adopt the classical Limited Discrepancy Search (LDS) algorithm introduced by Harvey & Ginsberg (1995) to optimize the pipeline selection process. This algorithm is essentially a depth-first search strategy that searches for new set of solutions by iteratively increasing the number of discrepancy values where the discrepancy refers to the number of variables in the current configuration that differ from their values in the initial configuration. The novel contributions of this paper include:

- Distributed autotuning of LoRA configurations for pre-trained transformer models to find the optimal pipeline for the target domain. To the best of our knowledge, this is the first paper to explore the potential of autotuning time series transformer models.

- The adoption of LDS for exploring the LoRA hyper-parameter search space in autotuning to minimize computational overhead.

- Extensive tests across a suite of out-of-domain benchmark datasets to compare the performance of autotuned transformer models, traditional fine-tuning strategies and zero shot pre-trained transformer models.

## 2 RELATED WORK

There has been a rapid surge in the transformer-based time series forecasting techniques recently, particularly for long-term forecasting. We briefly introduce some of these transformers in this section. The authors in Zhou et al. (2021) propose a transformer-based model named Informer to employ a probability sparse attention mechanism to capture long-term dependencies in time series data. The ProbSparse self-attention mechanism along with the distilling operation is used to handle the challenges of quadratic time complexity and high memory usage in the vanilla Transformer architecture. Then, the authors of Autoformer Wu et al. (2021) introduced a decomposition transformer architecture replacing the attention module with an AutoCorrelation mechanism to gain progressive decomposition capacities outperforming self-attention in terms of both efficiency and accuracy. To further reduce the computational cost of transformer from quadratic to linear, FEDformer Zhou et al. (2022), a Frequency Enhanced Decomposed Transformer was introduced. It captured the important global/overall structures in time series by means of frequency domain mapping achieving linear complexity. PatchTST Nie et al. (2022) on the other hand, introduced two key components : segmentation of time series into patches serving as input tokens to the transformer along with channel-independence. Each channel contains a single univariate time series sharing the same embedding and Transformer weights across all the series for long-term multivariate time series forecasting and self-supervised representation learning. This overcomes the challenge of computation and memory usage of attention maps enabling the model to benefit from longer look-back windows. Yet another pioneering work in this area is Lag-Llama Rasul et al. (2023) inspired by the LLaMA Touvron et al. (2023) LLM. This model utilizes a simple decoder-only transformer architecture for time series forecasting by using lagged features as covariates. The main problem addressed by this work is the application of foundation model approach to time series data and investigation of the extent of the transfer achievable across a wide range of time series domains. Another line of work which treats time series as strings is LLMTime Gruver et al. (2024a). They employ careful

pre-processing specific to the given LLMs' tokenizer, demonstrating that LLMs can inherently perform zero shot forecasting. This is achieved by relying on LLM's abilities to extrapolate patterns in general sequences. Likewise, Chronos Ansari et al. (2024) employs the encoder-decoder transformer architecture from the T5 Raffel et al. (2020) LLM family requiring minimal modifications i.e, tokenization though scaling and quantization.

There has been a growing interest in the research field of AutoML in the recent past fuelled by the exponential growth in the availability of computational resources Khan et al. (2021). Moreover, it assists practitioners in developing accurate and efficient predictive models without extensive domain knowledge Feurer et al. (2015) Katz et al. (2016) Jin et al. (2019). Hyperparameter optimization (HPO) acts as an important component of AutoML in searching for optimal hyper-parameters for neural network structures and the model training process. There is a strong need to adapt these pre-trained time series transformer models for specific downstream forecasting tasks for improved performance in the target domain Liang et al. (2024). Leveraging AutoML for this process of fine-tuning can help achieve this goal by automating the task of fine-tuning in a computationally feasible manner. Most commonly, PEFT techniques have been proposed in NLP and CV for fine-tuning a subset of parameters in various downstream tasks. These are mainly classified into two groups: selective and additive. Selective PEFT approaches involve fine-tuning a selective set of parameters in the model architecture as shown in Touvron et al. (2022) tuning only the attention layers. Unlike selective approaches, additive PEFT adds new weights into the model known as adapter modules and fine-tune only these, keeping the existing weights frozen. One such popular adapter is LoRA Hu et al. (2021a) which adds trainable low-rank matrices into transformer layers to approximate the weight updates. The application of LoRA in the domain of fine-tuning time series transformers remains largely unexplored due to the rapidly evolving nature of this field and hence, motivates this work. Moreover, to further optimize the search space for hyperparameter optimization in the AutoML pipeline, we adopt LDS into our framework to find the best performing model configuration.

In contrast to the transformer architectures and algorithms considered in this related work, we focus on the efficient autotuning of pre-trained time series transformers for a given target domain.

## 3 METHODOLOGY

In this section, we first define the problem statement and then present a detailed description of the autotune design approach we adopted using LoRA and LDS.

**Problem Statement**: Given a univariate time series dataset $X = (x_0, x_1, .., x_n)$ where $x_n \in \mathbb{R}$ is a time ordered sequence of real values and $n$ is the length of the time series and given context length $c$ where $1 < c < n$, we used the time series $x_{1:c} = x_1, x_2, \ldots, x_c$ to forecast the time series $x_{c:c+h} = x_{c+1}, x_{c+2}, \ldots, x_{c+h}$, where $h$ is the forecast horizon and $c + h = n$.

To solve the forecasting task described above, we used Chronos T5 models as the time series transformers which have been pre-trained on a large collection of publicly available time series datasets from varied domains. However, since these models are trained on a broad spectrum of time series data, their performance on a specific task which in our case is the unseen target dataset may not be optimal.

**Autotune using LoRA and LDS**: We design a novel algorithm to perform automated fine-tuning of transformer models using LoRA as the parameter efficient tuning method in conjunction with LDS. This fine-tuning is achieved by using a distributed ray based framework. Figure 1 shows the architecture diagram of our approach wherein we take a pre-trained Chronos T5 model and specify a minimal subset of weights to be trained to adapt the model on the target dataset using LoRA. LoRA Hu et al. (2021a) implements this in a storage and compute-efficient manner by constraining the update ($\Delta W$) to the pre-trained weight matrix $W_0 \in \mathbb{R}^{d \times k}$ by representing it with a low-rank decomposition,

$$W_0 + \Delta W = W_0 + BA$$

where $B \in \mathbb{R}^{d \times r}$, $A \in \mathbb{R}^{r \times k}$, and the rank $r \leq \min(d, k)$. During fine-tuning, $W_0$ is frozen, while the weights of $A$ and $B$ are updated. We adapt the weight matrices corresponding to the self-attention module and the feed-forward layer modules of the transformer architecture.

Algorithm 1 outlines the steps involved in our autotune approach. The algorithm starts with initialising the LoRA hyper-parameter search space. We use Limited Discrepancy Search or

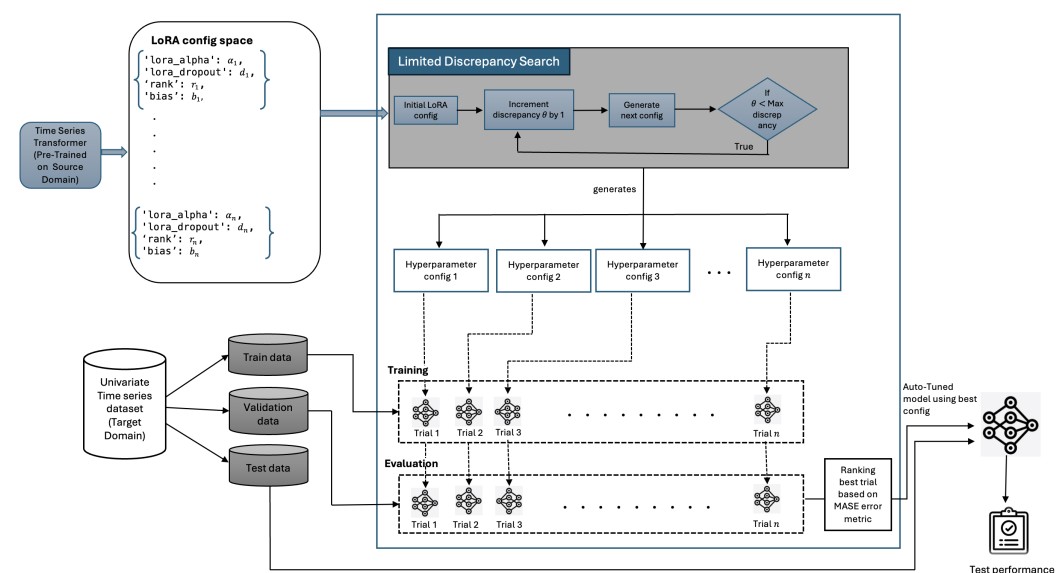

Figure 1: Architecture diagram showing the workflow of our algorithm using LoRA and LDS for Hyperparameter Optimization and Ray Tune for parallelization of the tune trials.

---

**Algorithm 1** Autotune algorithm using LoRA and LDS

---

**Input:** Model $M$, target dataset $X$ split into $X_{\text{train}}$, $X_{\text{val}}$ and $X_{\text{test}}$, LoRA search space $\mathbf{Y} = \{Y_1, ..., Y_n\}$, SEARCH with maximum discrepancy $\delta$, Evaluation metric $MASE$, number of trials $T_{max}$

**Output:** Optimal LoRA hyper-parameters $Y_{opt}$ and Autotuned Model $M_{Y_{opt}}$

1: Define LoRA search space $\mathbf{Y} = \{Y_1, ..., Y_n\}$
2: Execute SEARCH : Initialize $\mathbf{y^0}$ to default LoRA and let $\mathbf{y}^* \leftarrow \mathbf{y^0}$
3: **for** all $\theta = 1, ..., \delta$ **do**
4:     SEARCH($\mathbf{y}^*, \mathbf{Y}, \theta, 1$)
5: **end for**
6: **return** $\mathbf{y}^*$
7: **procedure** SEARCH($\mathbf{y}, \mathbf{Y}, \theta, i$)
8:     **if** $\theta == 0$ **or** $i > |\mathbf{Y}|$ **then**
9:         $Y_{opt}, M_{Y_{opt}} \leftarrow$ SCORE($\mathbf{y}, X_{\text{train}}, X_{\text{val}}, M$)
10:         **return** $Y_{opt}, M_{Y_{opt}}$
11:     **else**
12:         **for** all values $y \in D(\mathbf{Y}[i])$ **do**

13:             **if** $\mathbf{y}[i] == y$ **then**
14:                 $z \leftarrow$ SEARCH($\mathbf{y}, \mathbf{Y}, i + 1, \theta$)
15:             **else**
16:                 $\mathbf{y}' \leftarrow \mathbf{y}$; $\mathbf{y}'[i] \leftarrow y$
17:                 $z \leftarrow$ SEARCH($\mathbf{y}', \mathbf{Y}, i + 1, \theta - 1$)
18:             **end if**
19:             **return** $z$
20:         **end for**
21:     **end if**
22: **end procedure**
23: **procedure** SCORE($\mathbf{y}, X_{\text{train}}, X_{\text{val}}, M$)
24:     $M \leftarrow$ TrainModel($\mathbf{y}^*, X_{\text{train}}$)
25:     score $\leftarrow$ EvaluateModel($M, X_{\text{val}}$)
26:     **if** score $>$ best_score **then**
27:         $M_{Y_{opt}} \leftarrow M$
28:         $Y_{opt} \leftarrow \mathbf{y}^*$
29:         best_score $\leftarrow$ score
30:     **end if**
31:     **return** $Y_{opt}, M_{Y_{opt}}$
32: **end procedure**

---

LDS Harvey & Ginsberg (1995) to traverse this space effectively starting from an initial configuration of LoRA hyper-parameters. Specifically, LDS takes as input a vector of variables $\mathbf{Y} = \{Y_1, ..., Y_n\}$ corresponding to the LoRA hyper-parameters together with their domains of values $\mathbf{D} = \{D(Y_1), ..., D(Y_n)\}$ representing the hyper-parameter search space to be explored and the maximum discrepancy value $\delta$ which limits the number of allowed variable-value assignment changes from the initial solution $\mathbf{y}^0 = (y_1^0, ..., y_n^0)$ where $y_i^0 \in D(Y_i)$ is a value in variable's $Y_i$ domain and outputs the next solution $\mathbf{y}^*$ based on the discrepancy value. Notice that LDS requires

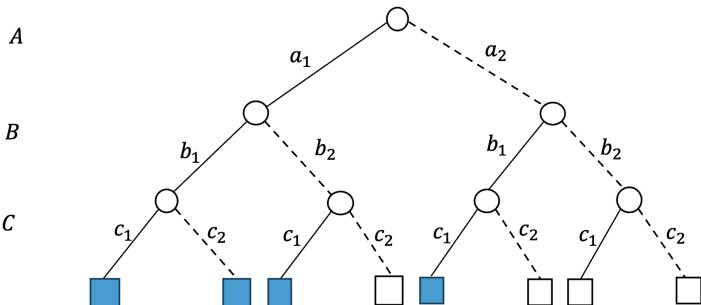

Figure 2: Example of the search space traversed by LDS with maximum discrepancy value = 1.

the variables to have finite domains of values, and therefore any continuous hyper-parameter needs to be discretized. In addition, LDS is required to search around a reasonably good solution which is typically given by the default LoRA hyper-parameter configuration here, hence, we set $\mathbf{y}^0$ to the default LoRA configuration. We begin with a discrepancy value $\theta$ of 1 and conduct an iterative search that allows to change the values of at most $\theta$ hyper-parameters in the initial solution $\mathbf{y}^0$. We then increment $\theta$ until $\theta$ exceeds the maximum discrepancy value. Function SEARCH performs the actual exploration of the LoRA hyper-parameter search space limited by discrepancy $\theta$. This assists in incrementally searching around the default LoRA configuration compared to random exploration. For each configuration $\mathbf{y}'$ returned by LDS, we fine-tune the model and evaluate it on the validation split to compute the MASE score as outlined in the SCORE function in the algorithm. At the end, we find the best configuration $\mathbf{y}^*$ corresponding to the lowest MASE score and use it to fine-tune our model which is then evaluated on the held out test split.

For illustration, Figure 2 shows the search space explored by LDS with discrepancy value 1 (denoted by LDS(1)) for 3 dummy variables [A, B, C] with domain values $\{a_1, a_2\}$, $\{b_1, b_2\}$ and $\{c_1, c_2\}$, respectively. In this case, LDS(1) starts from the initial assignment of $\{a_1, b_1, c_1\}$ which corresponds to the leftmost blue leaf node and traverses the search space in a depth-first manner visiting only the blue leaf nodes.

The algorithm described above is then implemented in a distributed manner using Ray Tune Liaw et al. (2018) which provides an open source framework for distributed model training and selection. Each configuration returned by LDS corresponds to trials as shown in Figure 1 which are executed concurrently in a cluster. We use the default Ray tune scheduler which is first-in-first-out (FIFO) passing through the trial configurations without performing any early stopping.

## 4 EXPERIMENTAL SETUP

In this section, we present the datasets used in the fine-tuning experiments along with the implementation details of the proposed autotune framework.

**Datasets**: For our experiments, we use 10 datasets from the Monash Time Series Forecasting Repository Godahewa et al. (2021) part of the Benchmark II datasets in Ansari et al. (2024) used for zero shot evaluation. Table 1 provides the details of the datasets used for the experiments. Each dataset is a collection of series where each series is split into train, validation and test instances. The * in the series length represents average length of the time series for 3 datasets, Weather, Australian Electricity and M5 which consists of variable length time series. The number of instances in validation and test label depend on the prediction horizon which is specific for each dataset. We use the GluonTS library to split the datasets using the same strategy as used in Ansari et al. (2024). For each dataset, the train split includes all the data points from the beginning of the time series up until the last two prediction horizon lengths, which are held out for validation and testing respectively. We use these datasets as they have not been used in the pre-training phase of the Chronos T5 models. Since these datasets are collected from a variety of application domains including energy, transport, retail, web, weather, finance, they are more closely representative of the real world application scenarios. Moreover, they assist in validating the robustness and generalisation capabilities of our approach.

Table 1: Datasets used for the experiments.

| Dataset | Domain | Freq. | Num. Series | Series Length | Prediction Length (H) |
|---|---|---|---|---|---|
| Traffic | Transport | 1H | 862 | 17544 | 24 |
| Weather | Nature | 1D | 3010 | 14296* | 30 |
| ETT (Hourly) | Energy | 1H | 14 | 17420 | 24 |
| ERCOT Load | Energy | 1H | 8 | 154854 | 24 |
| Australian Electricity | Energy | 30min | 5 | 231052* | 48 |
| Exchange Rate | Finance | 1B | 8 | 7588 | 30 |
| FRED-MD | Economics | 1M | 107 | 728 | 12 |
| NN5 (Daily) | Finance | 1D | 111 | 791 | 56 |
| M5 | Retail | 1D | 30490 | 1562* | 28 |
| ETT (15 min.) | Energy | 15min | 14 | 69680 | 24 |

**Models**: We chose Chronos T5 models as our transformer architecture for demonstrating the auto-tune approach. These models, trained from scratch on time series data, are widely adopted open-source transformers that have achieved state-of-the-art (SOTA) performance in time series analysis. Chronos T5 models have been pre-trained and released in 5 sizes ranging from Tiny (16M), Mini (20M), Small (46M), Base (200M) and Large (710M) with number of model parameters in brackets. In our experiments, we focus on univariate time series forecasting as Chronos models are pre-trained for the univariate setting. We use the lightweight version of the models i.e. Mini in order to utilize minimal computational resources for demonstrating the applicability of our approach.

Table 2: LoRa Hyper-paramater Search Space

| Parameter Name | Range of values |
|---|---|
| alpha | {4, 8, 16, 32, 64} |
| dropout | {0.0, 0.05, 0.1} |
| rank | {2, 4, 8, 16, 32} |
| bias | {"none", "all", "lora_only"} |
| learning_rate | {0.0001, 0.001, 0.01} |
| batch_size | {4, 8, 16} |
| grad_accumulation_steps | {1, 4, 8} |

**Implementation Details**: Autotune has been implemented using Ray Tune and Transformers library. It supports both encoder-decoder (seq2seq) and decoder-only (causal) models as well as parameter efficient fine-tuning (backed by the PEFT Library). Table 2 shows the search space for the LoRA hyper-parameters used in auto fine-tuning the models. We execute 10 trials selected using LDS for each dataset to find the best LoRA configuration and output the auto fine-tuned model corresponding to it. We limit the number of trials to 10 to demonstrate the robustness of our approach in a resource-constrained environment. We vary the maximum discrepancy based on the number of LoRA hyper-parameters to be tuned which in our case is equal to 8. Therefore, we experiment with maximum discrepancy values of 4 and 8 to explore the search space. A lower value of 4 involves a more focused search while 8 allows more broader exploration of the potential hyper-parameters. To ensure a comprehensive evaluation across different fine-tuning settings, we use mean absolute squared error (MASE) as the evaluation metric. Since the model produces probabilistic forecasts, the forecasted value for each datapoint is calculated as the median (0.5-quantile) of 20 samples, which is then used to calculate the metrics similar to Ansari et al. (2024). The MASE scores are averaged across 5 runs. All the experiments are performed on Macbook Pro M3 Max with 64GB RAM. We also perform full fine-tuning of the Chronos mini model described in Ansari et al. (2024) to compare the performance efficiency of our approach over fine-tuning all the model weights.

Table 3: MASE scores obtained using Chronos T5 mini model in Zero Shot, Full Fine-Tuning and Autotune using LoRa and LDS respectively.

| Dataset | Zero Shot | Full Fine Tuning | Autotune with LoRA and LDS |
|---|---|---|---|
| Traffic | 0.853 ($\pm$0.0012) | **0.727** ($\pm$0.0063) | 0.746 ($\pm$0.0173) |
| Weather | 0.859 ($\pm$0.0032) | **0.818** ($\pm$0.0046) | 0.821 ($\pm$0.0048) |
| ETT (Hourly) | 0.795 ($\pm$0.0111) | **0.783** ($\pm$0.0166) | 0.796 ($\pm$0.0260) |
| ERCOT Load | 0.582 ($\pm$0.0107) | 0.599 ($\pm$0.0352) | **0.565** ($\pm$0.0381) |
| Australian Electricity | 0.965 ($\pm$0.0406) | 0.927 ($\pm$0.0784) | **0.831** ($\pm$0.0923) |
| Exchange Rate | 2.054 ($\pm$0.1561) | 1.846 ($\pm$0.1178) | **1.631** ($\pm$0.1963) |
| FRED-MD | **0.473** ($\pm$0.0105) | 0.510 ($\pm$0.0151) | 0.510 ($\pm$0.0092) |
| NN5 (Daily) | 0.648 ($\pm$0.0059) | **0.603** ($\pm$0.0032) | 0.619 ($\pm$0.0156) |
| M5 | 0.942 ($\pm$0.0004) | 0.934 ($\pm$0.0003) | **0.925** ($\pm$0.0005) |
| ETT (15 min.) | **0.709** ($\pm$0.0269) | 0.777 ($\pm$0.0161) | 0.713 ($\pm$0.0427) |

Table 4: Comparison of MASE scores obtained using Zero Shot evaluation of Chronos T5 models across different sizes with Autotune using LoRa and LDS.

| Dataset | Zero Shot | | | | Autotune with LoRA and LDS |
|---|---|---|---|---|---|
| | Large | Base | Small | Mini | |
| Traffic | 0.795 | 0.800 | 0.821 | 0.853 | **0.746** |
| Weather | 0.817 | **0.815** | 0.854 | 0.859 | 0.821 |
| ETT (Hourly) | 0.768 | **0.757** | 0.792 | 0.795 | 0.796 |
| ERCOT Load | 0.586 | **0.501** | 0.597 | 0.582 | 0.565 |
| Australian Electricity | 1.411 | 1.185 | 1.256 | 0.965 | **0.831** |
| Exchange Rate | 2.214 | 2.466 | 2.015 | 2.054 | **1.631** |
| FRED-MD | 0.516 | 0.483 | **0.473** | 0.473 | 0.510 |
| NN5 (Daily) | **0.576** | 0.593 | 0.615 | 0.648 | 0.619 |
| M5 | 0.946 | 0.942 | 0.936 | 0.942 | **0.925** |
| ETT (15 min.) | 0.731 | **0.661** | 0.740 | 0.709 | 0.713 |

## 5 RESULTS

We present our comparative results in Table 3 which shows the performance of Chronos mini T5 model in zero shot setting along with different fine-tuning settings. We report MASE averaged over 5 runs for both zero shot and full fine-tuning setting. For our approach, we run 10 trials and report MASE corresponding to the best LoRA configuration, which is also averaged over 5 runs. We observe that the performance of our approach is better than full fine-tuning for most of the datasets except for datasets in the domain of traffic, weather and electricity where full fine-tuning overtakes by a slight margin. This can be attributed to the fact that the pre-trained Chronos T5 model has seen datasets from the aforementioned domains during the pre-training phase as mentioned in Ansari et al. (2024). However, for target domain datasets such as exchange rate and M5 which do not share any similarity with the pre-training source datasets, our approach demonstrates the best performance. This validates our claim that LoRA achieves comparable, and often superior performance while significantly reducing the number of trained parameters compared to full fine-tuning for out-of-domain target datasets. In Figure 3, we also visualize the average MASE improvement percentage achieved by our approach compared to zero shot model across 10 datasets. On an average, there is a 5.21% improvement denoted by the red dotted line in the plot. We can see that the autotuned model outperforms the zero shot model for most datasets. Moreover, it exhibits particularly strong performance on datasets that are out-of-domain for the original pre-trained model such as exchange rate with a significant MASE improvement of 20.59%.

Figure 4 shows the relative performance of the different fine-tuning methods across 10 datasets. This plot visualizes the performance of each method relative to the best-performing method for each

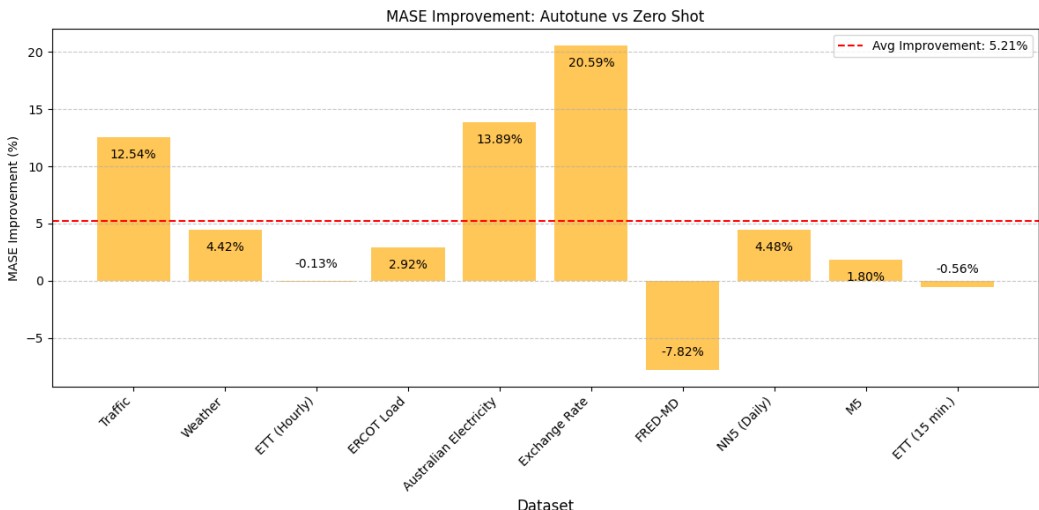

Figure 3: Percentage improvement obtained by autotuned mini model over zero shot mini model.

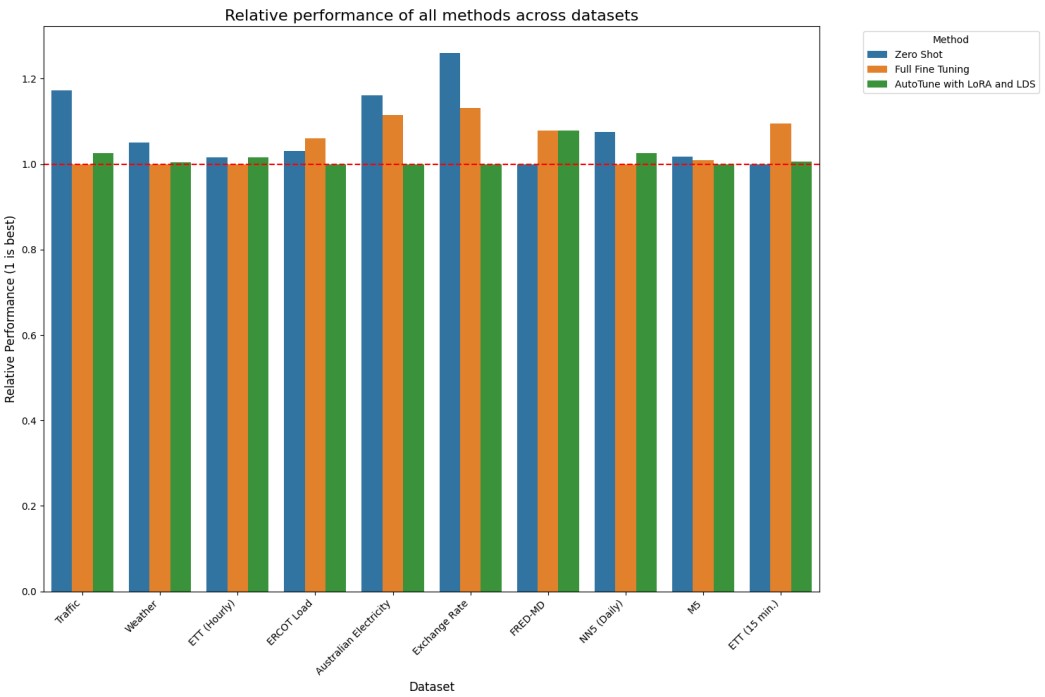

Figure 4: Plot showing the relative performance of all the methods across datasets.

dataset. It shows how close each method is to the best performing method for each dataset. A value of 1 indicates the best performance and is represented by the red dotted line. The longer the bar exceeds 1, the worse the performance becomes. It can be clearly seen that autotune achieves the best performance for majority of the datasets indicated by the green bar coinciding with the red dotted line. However, in cases where autotuned model is not the best performing, it only misses by a slight margin.

Next, we also compare the performance of our autotuned mini model against the zero shot performance of different sizes of the Chronos T5 models to highlight the potential of fine-tuned smaller models which require fewer computational resources compared to their larger counterparts. Table

4 shows the performance of the autotuned model in comparison to the zero shot results obtained using different sizes of the Chronos T5 models namely: Large, Base, Small and Mini. Here, we observe that our autotuned mini model outperforms the zero shot mini models for all the datasets with an exception of 2 datasets namely: FRED-MD and ETT(15min). This shows that fine-tuning using LoRA and LDS helps in efficiently adapting the model to the target domain than directly using zero shot model in 8 out of 10 datasets. Another interesting finding that is clearly evident in Figure 5 where we visualise the results from Table 4 is that our autotuned mini model even outperforms the zero shot Chronos T5 small model for 6 out of 10 datasets including Traffic, Weather, ERCOT Load, Australian Electricity, Exchange Rate and M5. Moreover, it surpasses the performance of the zero shot large model for 3 of the above 6 datasets which potentially leads to significant cost savings given the huge difference in the model sizes.

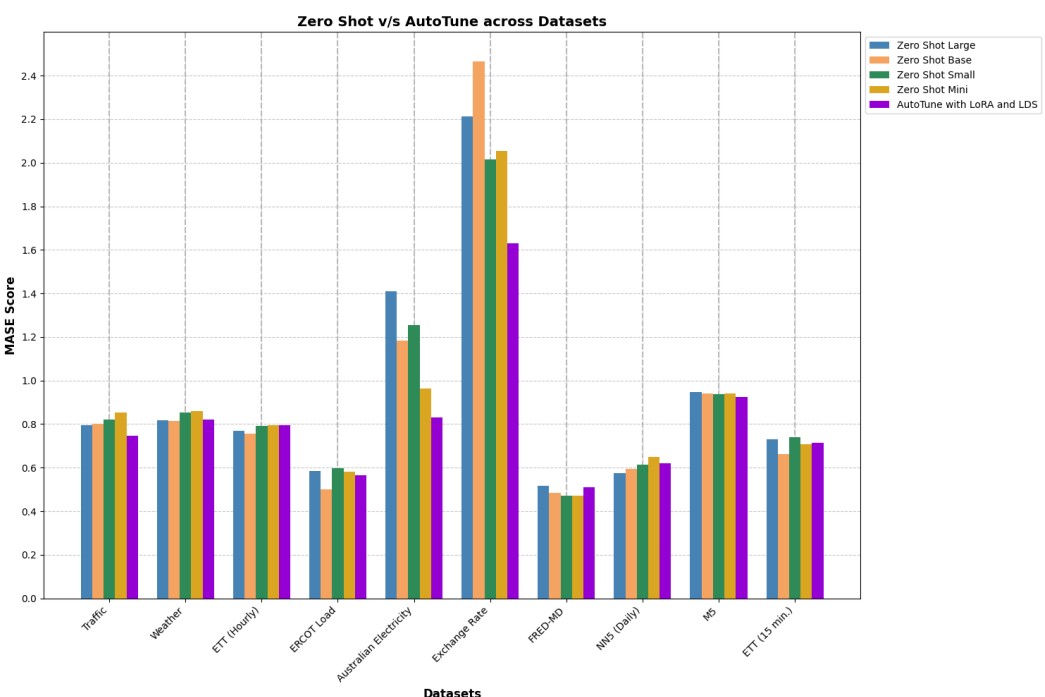

Figure 5: Plot showing the comparison of MASE scores obtained using Autotune with Zero shot models of different sizes.

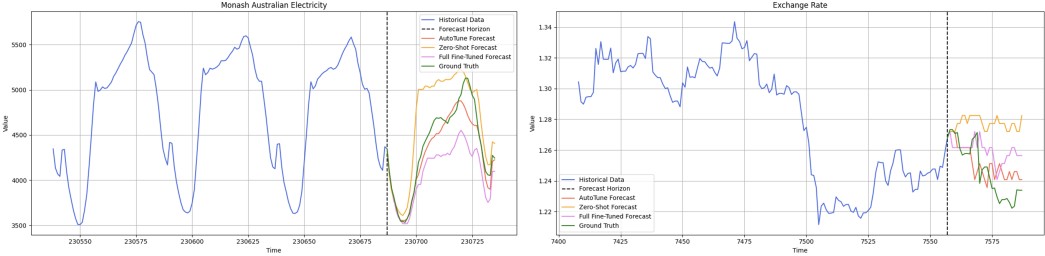

Figure 6: Performance comparison for Zero shot, Full fine-tuned and Autotuned models for Monash Australian Electricity and Exchange Rate datasets.

Figure 6 illustrates the predictive performance of the various models on the target domain datasets : Monash Australian Electricity and Exchange Rate respectively. Here, we compare the forecasts obtained on the test split by our autotuned model with the other models, clearly highlighting the improved prediction accuracy obtained using our approach. In summary, our findings demonstrate that our approach can efficiently autotune time series transformers by searching the most optimal hyper-parameters to enhance their downstream performance in the target domain.

## 6 CONCLUSION AND FUTURE WORK

Time series transformers efficiently capture long-range patterns and dependencies, improving the model's ability to predict complex temporal relationships. However, their successful application to specific downstream tasks needs adaptation to the target domain datasets which can be achieved via fine-tuning. In this work, we propose autotuning time series transformers using parameter efficient fine-tuning method LoRA along with LDS as the search strategy. Our approach outperforms full fine-tuning specifically for out-of-domain datasets not seen during the model pre-training phase. Furthermore, we show that our autotuned model also beats the zero shot mini models for 80% of the benchmark datasets surpassing even the performance of the zero shot large models in some cases. Furthermore, since our autotuning approach is based on LoRA, it can be easily extended to other time series foundation models. In the future, we will also extend our autotune approach for multivariate time series datasets.

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
