# OpenReview forum: "AutoTune for Time Series Transformers using Low Rank Adaptation and Limited Discrepancy Search"
_ICLR.cc/2025/Conference — Submitted to ICLR 2025_

### Official Review · Reviewer_rofx · 2024-10-29

**Soundness:** 2
**Presentation:** 1
**Contribution:** 1
**Rating:** 1
**Confidence:** 5

**Summary:**

This paper covers the application of Limited Discrepancy Search (LDS) to optimize LoRA finetuning for transformer-based time-series forecasting models (Chronos in this case). The core idea is to optimize the LoRA training hyperparameters to automate the finetuning of time-series forecasting models for downstream tasks. The authors present results on several different datasets from different domains. They fix the hyper-parameter search space so that each parameter has a finite set of values. They then compare their results versus the zero-shot chronos model, the fully finetuned version and the autotuned version. They show that on average their method increases performance relative to the other versions.

**Strengths:**

1.	Auto-tuning of time-series models is relatively understudied
2.	They perform their tests on several datasets from a wide range of domains
3.	They choose a model architecture which comes in a range of sizes which makes the results more interesting

**Weaknesses:**

1.	The novelty is lacking. This research takes a pretrained model and applies LoRA finetuning to it with a new method of hyperparameter tuning. Most of these concepts are not new for time-series analysis
2.	The paper claims to increase performance but often this strategy does not improve the quality of the forecasting. Given that you are finetuning the model for a downstream task how is it that the performance is getting worse in some cases (Table 3 and 4)?
3.	You claim the autotuned model method is improving the performance but what if you simply finetune the models with LoRA and a standard set of hyperparameters would the performance be the same as your method? This is a question you need to answer clearly.
4.	Why do you need figure 5 and table 4? Its looks like it is showing the same thing, it looks like you were trying to fill the space. Figure 4 also looks unnecessarily large and is a generally convoluted way of showing the results.
5.	Why would you not apply this method to each model size, why is it only applied once?
6.	The tuning strategy only seems to be applied to the mini model, a key advantage of LoRA is that you can train a larger model with less compute. Which is the opposite of what was chosen here.
7.	Why are there standard deviations in table 3 but not in table 4.
8.	Figure 2 is not centered
9.	Some references should be in parathesis (for example line 38).
10.	All of these models tested are relatively small and could be trained on somewhat accessible hardware even for academia. Why apply LoRA in this case?

**Questions:**

In general ,I think this paper is too low impact for this conference. Their strategy lacks novelty and the results are not rigorous enough. Particularly in the validation of their method. I do have some questions pertaining to the weaknesses mentioned above:
1. Why did you not compare to a model finetuned using LoRA with default hyperparameters?
2. Can you show how this works with other models? There are many other pretrained time-series forecasting models you could do this with.
3. How does this method improve the accuracy over other existing hyperparameter searching methods?
4. Finetuning a model for time-series analysis with LoRA is not novel, performing a hyperparameter search for the LoRA hyperparameters is not novel, how do we know that the performance increases are due to the use of the LDS algorithm with LoRA?

---

> ### Author Response · Authors · 2024-11-25
>
> We sincerely appreciate your detailed and thoughtful suggestions. Please find our comments on each of the questions in detail.
>
> **Question 1. Why did you not compare to a model finetuned using LoRA with default hyperparameters?**
>
> Answer 1. Since default LoRA hyperparameters is the initial configuration that LDS starts with, it automatically is being considered in the ranking of models on the validation set. Therefore, in the case where the default LoRa is the best configuration then Autotune will select that and return the atotuned model fine tuned using the default LoRa configuration. It is observed that for two datasets, ERCOT Load and NN5(Daily) Autotuned model is fine-tuned using default LoRA hyperparameters.
>
> **Question 2. Can you show how this works with other models? There are many other pretrained time-series forecasting models you could do this with.**
>
> Answer 2. Yes, this method can be applied to other models. Our initial experiments with TinyTimeMixer (TTM) models which handle multivariate time series demonstrate the effectiveness of Autotune on two datasets : ETTh1 and weather. The percentage improvement in RMSE score gained using Autotune for ETTh1 and weather is reported as 0.4% and 6.07% respectively compared with zero-shot performance. These preliminary results reiterate our findings that Autotune helps in improving the model performance in the target domain.
>
> **Question 3 : How does this method improve the accuracy over other existing hyperparameter searching methods?**
>
> Answer 3. The rational behind using LDS for hyperparameter searching over other naive methods like random/grid search is that  random search does not guarantee the best solution and there will always be an element of randomness to whether the best hyperparameters will be found or not.
>
> For grid search, it will explore the space starting from the initial solution exhaustively searching for the best hyperparmaters. This can prove to be extremely computationally expensive given the time complexity of fine-tuning Time Series Foundation Models. LDS, on the other hand,  gives us the freedom to control the search space exploration using the maximum discrepancy value which limits how narrow or broad we want to go into the search space. This cannot be achieved by using random/grid search. We would agree that grid search would eventually converge to the most optimal set of hyperparameters but at the cost of very high computational time (days or weeks depending on model and dataset size) given the number of combinations possible for the specified hyperparameter search space.  Moreover, it has been demonstrated that LDS returns an optimal set of hyperparameters even when we only explore 10 possible configurations corresponding to 10 trials. The results demonstrate a  significant improvement in the MASE scores specifically for out of domain datasets.
>
> Other more popular hyperparameter search methods like Bayesian Optimization (BO) were not considered in this work and can be an interesting future direction to investigate the process of further optimising the Autotune pipeline.
>
>
> **Question 4 : Finetuning a model for time-series analysis with LoRA is not novel, performing a hyperparameter search for the LoRA hyperparameters is not novel, how do we know that the performance increases are due to the use of the LDS algorithm with LoRA?**
>
> Answer 4. We agree that fine-tuning or auto-tuning is not a novel concept in itself, however, it has not been explored in the context of Time Series Forecasting using Foundation Models. The only work that uses LoRA for domain adaptation using time series models is limited in its study to using default LoRA on private medical data [1]. Our work uses Chronos models to demonstrate the effectiveness of auto fine-tuning and the results on open source benchmark datasets emphasise the need of Autotune in this rapidly evolving space of Time Series Foundation models.
>
> **References**
>
> [1] Gupta, Divij, et al. "Low-Rank Adaptation of Time Series Foundational Models for Out-of-Domain Modality Forecasting." arXiv preprint arXiv:2405.10216 (2024).

---

> > ### Comment · Reviewer_rofx · 2024-11-25
> > **Reviewer Response**
> >
> > Thank you for responding to some of my questions.
> >
> > However, I am still not convinced about the overall quality of the paper. Using LoRA finetuning to improve the model accuracy for downstream tasks is not a surprising result and the methods are not rigorous enough to validate the use of LDS. Most importantly the authors state the advantages of LDS over some existing methods but do not have the experimental results to validate these claims. Additionally, most of the weaknesses mentioned above have not been addressed. There must be at least one experiment which compares the effectiveness of LDS to other searching strategies.
> >
> > I also believe that even with more comprehensive analysis, the novelty of the submission is concerning. I am  not entirely convinced that applying LoRA to foundation models for time-series forecasting is novel. For example both TEMPO [1] and TEST[2] employ parameter efficient finetuning methods for time-series forecasting.
> >
> > Content aside, the presentation quality of the paper is an issue for a venue of this caliber. Many of the plots are far too large and are not an efficient use of the space. The formatting between tables is not consistent (see weakness 7).
> >
> >
> > For these above reasoning I have decided to maintain my score. Thank you for responding to my review.
> >
> > ------
> >
> > [1] @inproceedings{
> > cao2024tempo,
> > title={{TEMPO}: Prompt-based Generative Pre-trained Transformer for Time Series Forecasting},
> > author={Defu Cao and Furong Jia and Sercan O Arik and Tomas Pfister and Yixiang Zheng and Wen Ye and Yan Liu},
> > booktitle={The Twelfth International Conference on Learning Representations},
> > year={2024},
> > url={https://openreview.net/forum?id=YH5w12OUuU}
> > }
> >
> > [2]@inproceedings{
> > sun2024test,
> > title={{TEST}: Text Prototype Aligned Embedding to Activate {LLM}'s Ability for Time Series},
> > author={Chenxi Sun and Hongyan Li and Yaliang Li and Shenda Hong},
> > booktitle={The Twelfth International Conference on Learning Representations},
> > year={2024},
> > url={https://openreview.net/forum?id=Tuh4nZVb0g}
> > }

---

### Official Review · Reviewer_pYcv · 2024-11-02

**Soundness:** 2
**Presentation:** 2
**Contribution:** 2
**Rating:** 3
**Confidence:** 3

**Summary:**

This paper proposes a new method for autotune time series transformers, combining Low Rank Adaptation (LoRA) and Limited Discrepancy Search (LDS) to efficiently perform parameter optimization on pre-trained time series models. In this paper, LoRA is used for efficient parameter tuning, and LDS is combined to explore the optimal hyperparameter configuration to make the model perform better in the target domain. Experimental results show that compared with the zero-shot pre-trained model and the traditional full-parameter fine-tuning, the proposed method achieves better performance on multiple datasets.

**Strengths:**

1. This paper proposes an innovative automatic tuning method that combines the efficient parameter fine-tuning of LoRA and the search strategy of LDS to solve the problem of adaptability of large-scale models in time series.
2. Experimental results show that the proposed method has significant performance improvement on multiple datasets, especially on some target domain datasets, which has significant advantages over zero-shot models.
3. In the process of LoRA fine-tuning, only a small number of parameters need to be adjusted, which significantly reduces the requirements of computing resources compared with full-parameter fine-tuning, thus greatly saving the computing cost.
4. This paper shows that the method has achieved good results on a variety of target domain datasets, showing the versatility and adaptability of the method.

**Weaknesses:**

1. Although LoRA is suitable for efficient parameter fine-tuning, its application is mainly concentrated in the case of large differences between the target domain and the source domain, and the effect improvement in some specific fields is limited.
2. Although LDS optimizes the search space to a certain extent, its essence is still a depth-first search based on limited differences, and the search efficiency may be limited in the face of a larger hyperparameter space.
3. The use of LDS for hyperparameter searching in LoRA is a key innovation presented in this paper; however, the article does not provide a detailed experimental analysis of this method. It remains unclear whether LDS leads to a reduction in search iterations or an improvement in search efficiency compared with other hyperparameter search methods.
4. This paper employs LoRA as a fine-tuning method for time series forecasting models, demonstrating competitive results compared to zero-shot and full fine-tuning methods. However, as LoRA is already established as a general parameter-efficient fine-tuning approach, such results are widely evidenced in the literature，which, consequently, diminishes the contribution of this paper.
5. Although the paper compares the zero-shot model and the full-parameter fine-tuning, it does not make an in-depth comparison of other advanced fine-tuning methods, such as Adapter or other AutoML strategies, which limits the comprehensiveness of the comparison results.

**Questions:**

I have some doubts about the specific implementation details of the combination of LDS and LoRA in the paper. For example, the selection criteria for LDS and how to effectively handle the size of the hyperparameter space, and whether tuning on different datasets will be affected by specific hyperparameters. In addition, there is a clear definition of the "optimal" hyperparameters in the experimental setup, which may affect the interpretation of the results.

---

> ### Author Response · Authors · 2024-11-25
>
> We highly appreciate your detailed feedback on our work. Please find our comments aimed to address your questions in detail.
>
> **Questions**
>
>  I have some doubts about the specific implementation details of the combination of LDS and LoRA in the paper. For example, the selection criteria for LDS and how to effectively handle the size of the hyperparameter space, and whether tuning on different datasets will be affected by specific hyperparameters. In addition, there is a clear definition of the "optimal" hyperparameters in the experimental setup, which may affect the interpretation of the results.
>
> Answer:
>
> - LDS explores the search space relative to the initial configuration provided. The authors in LoRA paper [1] have come up with a default value of hyperparameters (rank, alpha, bias, dropout) which is a good reasonably good starting solution for most applications scenarios. We use this same set of default parameters for LDS as the starting solution and then control the selection of the next configuration using the maximum discrepancy value.
>
> - Yes, tuning on different datasets will be affected by specific hyperparameters like rank as stated in LoRA paper [1] “However, we do not expect a small r to work for every task or dataset. Consider the following thought experiment: if the downstream task were in a different language than the one used for pre-training, retraining the entire model (similar to LoRA with r = dmodel) could certainly outperform LoRA with a small r”.
>
> - Optimal hyperparameters in our experimental setup are determined based on the evaluation results on the validation split of the given dataset. In our case, hyperparameters are considered "optimal" if they minimize MASE on the validation dataset.
>
> **References:**
> [1] Hu, Edward J., et al. "Lora: Low-rank adaptation of large language models." arXiv preprint arXiv:2106.09685 (2021).

---

### Official Review · Reviewer_X838 · 2024-11-04

**Soundness:** 2
**Presentation:** 2
**Contribution:** 2
**Rating:** 5
**Confidence:** 3

**Summary:**

This paper proposes a novel method for automatic tuning of time series Transformer models, combining Low Rank Adaptation (LoRA) and Limited Discrepancy Search (LDS). The method aims to efficiently fine-tune pre-trained models, addressing the computational complexity associated with full parameter fine-tuning. LDS is used to optimize the hyperparameters of LoRA to enhance the model's adaptability to target domain tasks. Experimental results demonstrate that the proposed automatic tuning method outperforms both zero-shot and full fine-tuning approaches on most datasets, particularly in unseen target domains.

**Strengths:**

- The paper presents an efficient parameter tuning scheme for Transformer models in time series forecasting by combining Low Rank Adaptation (LoRA) with Limited Discrepancy Search (LDS).

- The effectiveness of Autotune is demonstrated through experiments on multiple datasets.

**Weaknesses:**

1. **Experimental Design Limitations**:
   - (1) The main experiments lack performance reports of Autotune on different sizes of the Chronos models. Although the authors stated that only the smallest model size was used to validate the proposed method's applicability, they also compared the performance of all sizes of Chronos T5 models under a zero-shot setting. Therefore, reporting the Autotune results for all model sizes would make the findings more convincing.
   - (2) The evaluation metric is singular. Although MASE reflects the overall performance improvement, it fails to capture other characteristics such as overfitting risks, error distribution, and extreme value prediction capabilities. The original Chronos-T5[1] also used WQL, and the work [2] employed additional metrics such as MSE and DTW.
   - (3) Missing ablation study. The authors used the LDS search algorithm to find the optimal LoRA hyperparameter settings, but there is no ablation study on the LDS algorithm itself. Including a comparison with the best hyperparameters selected after n random trials would help demonstrate the significance and necessity of the LDS algorithm.

2. **Limited Novelty**:
   - Although the authors claim that this is the first work to explore parameter-efficient fine-tuning in time series forecasting (Line 56), there are earlier studies that have explored this area (e.g., Low-Rank Adaptation of Time Series Foundational Models for Out-of-Domain Modality Forecasting). Furthermore, the effectiveness of the LDS search algorithm has not been sufficiently validated through ablation studies.

3. **Writing Issues**:
   - The citation format throughout the paper results in unclear and difficult-to-understand statements, such as those in Lines 38-42.

**Questions:**

1. Why was only one evaluation metric chosen? Are there any specific reasons related to the task setup for this decision?

2. Why were ablation studies and parameter analysis experiments not provided? Most of the figures in the experimental section only report the MASE scores compared to the baseline across different datasets, adequately demonstrating the effectiveness of the proposed method. However, would including ablation studies on the LDS search algorithm make the findings more persuasive?

3. How efficient is Autotune? After fine-tuning multiple hyperparameters and selecting the best-performing model on the validation set, is there a significant improvement compared to randomly selecting a set of hyperparameters (e.g., from Table 2) for a single LoRA fine-tuning?

4. Can the necessity of the LDS algorithm be demonstrated? The authors could provide a performance comparison by randomly selecting parameters from Table 2, training the model, and selecting the best-performing model on the validation set after n iterations.

---

> ### Author Response · Authors · 2024-11-25
>
> We would first like to thank all reviewers for their insightful comments on our work. We highly appreciate the valuable feedback on our paper, and are committed to addressing your concerns and removing any ambiguities.
>
> We will begin with answering each question specifically.
>
> **Question 1. Why was only one evaluation metric chosen? Are there any specific reasons related to the task setup for this decision?**
>
> Answer 1 : There is no specific reason related to the task setup here. MASE is a common evaluation metric used in the SOTA forecasting methods and other related works. Moreover, we choose this metric to be consistent with the Chronos paper which uses the same metric for evaluation of the models.
>
> **Question 2. Why were ablation studies and parameter analysis experiments not provided? Most of the figures in the experimental section only report the MASE scores compared to the baseline across different datasets, adequately demonstrating the effectiveness of the proposed method. However, would including ablation studies on the LDS search algorithm make the findings more persuasive?**
>
> Answer 2 : We acknowledge your feedback on the evaluation of LDS using ablation studies however, the main focus of our work was on the idea of Autotune working in principle for Time Series Foundation Models which has not been explored yet in the literature. Therefore, we did not include ablation studies.
>
> **Question 3. How efficient is Autotune? After fine-tuning multiple hyperparameters and selecting the best-performing model on the validation set, is there a significant improvement compared to randomly selecting a set of hyperparameters (e.g., from Table 2) for a single LoRA fine-tuning?**
>
> Answer 3 : Autotune is highly efficient in finding the best set of hyperparameters compared to a random combination of hyperparameters from Table 2. We start with an initial configuration as the default LoRA parameters. Only for 2 datasets (ERCOT Load and NN5 Daily) default LoRA was the best configuration returned by Autotune. As we highlight, default LoRA just gives us a reasonable starting configuration to explore the search space. So in principle, it can be considered as the first obvious/random choice for a single LoRA fine-tuning.
>
> **Question 4. Can the necessity of the LDS algorithm be demonstrated? The authors could provide a performance comparison by randomly selecting parameters from Table 2, training the model, and selecting the best-performing model on the validation set after n iterations.**
>
> Answer 4 : The main justification we choose LDS algorithm over random/grid search is given below :
>
> **Random Search** :  It does not guarantee the best solution and there will always be an element of randomness to whether the best hyperparameters will be found or not.
>
> **Grid Search** : Grid search will explore the space starting from the initial solution exhaustively searching for the best hyperparameters. This can prove to be extremely computationally expensive  given the context of complex Time Series Foundation Models we are dealing with. LDS on the other hand gives us the ability to control the search space exploration using the maximum discrepancy value which limits how narrow or broad we want to go into the search space. This cannot be achieved by using random/grid search. We would agree that grid search would eventually converge to the most optimal set of hyperparameters but at the cost of very high computational time (days or weeks depending on model and dataset size) given the number of combinations possible for the specified hyperparameter search space.  Moreover, it has been demonstrated that LDS returns an optimal set of hyperparameters even when we only explore 10 possible configurations corresponding to 10 trials. The results demonstrate a significant improvement in the MASE scores specifically for out of domain datasets.

---

### Meta-Review · Area_Chair_1cWk · 2024-12-19

**Metareview:**

(a) Summary of Scientific Claims and Findings:

The paper proposes an approach to optimize the fine-tuning of transformer-based time-series forecasting models by combining Limited Discrepancy Search (LDS) and Low-Rank Adaptation (LoRA). The goal is to automate the search for optimal LoRA hyperparameters, improving model performance. The approach is tested across multiple datasets, demonstrating that LDS-enhanced LoRA fine-tuning can minimize forecasting errors like MASE.

(b) Strengths:

1. Innovative Auto-Tuning Approach: The use of LDS for hyperparameter optimization in LoRA-based time-series models is a novel contribution.

2. Wide Application: The method is applied to datasets across various domains, showcasing its versatility.

3. Performance Gains: The autotuning approach shows promising results, particularly on specific datasets.

4. LoRA for Time-Series: Applying LoRA to time-series forecasting is a novel direction, building on its use in language models.

(c) Weaknesses and Gaps:

1. Limited Novelty: The method primarily combines existing techniques, lacking a sufficiently novel contribution.

2. Insufficient Experimental Comparison: The paper lacks comparison with other hyperparameter tuning methods like grid search, weakening the claims of LDS superiority.

3. Presentation Issues: Some figures and tables are poorly formatted or redundant, detracting from the clarity of the paper.

4. Unclear Impact of LDS: The results don't convincingly demonstrate that LDS outperforms simpler tuning methods.

5. Model Size: The application to small models undermines LoRA's potential for efficiently fine-tuning larger models.

6. Limited Hyperparameter Sensitivity Analysis: The paper lacks a detailed exploration of how hyperparameter tuning affects model performance.

7. Unclear Results: There's no clear comparison between autotuned models and those fine-tuned with standard hyperparameters.

(d) Reasons for Rejection:

1. Lack of Novelty: The paper's contribution is not sufficiently new, as it primarily repurposes existing methods without offering significant new insights.

2. Experimental Weakness: The absence of a thorough comparison with established hyperparameter tuning methods weakens the paper's claims.

3. Presentation Quality: Formatting and figure clarity issues detract from the overall presentation.

4. Unconvincing LDS Impact: The lack of strong empirical evidence showing LDS's advantages over simpler methods weakens the paper's core argument.

5. Insufficient Results Analysis: The unexplored application to small models and lack of direct comparison with other fine-tuning methods undermines the paper's impact.

In conclusion, the paper lacks sufficient novelty, experimental depth, and clarity, leading to a recommendation for rejection.

**Additional Comments On Reviewer Discussion:**

During the rebuttal period, the authors addressed several points raised by the reviewers:

1. Novelty of the Approach: One reviewer expressed concerns about the novelty of the proposed method, noting that it largely combines existing techniques like LoRA fine-tuning and hyperparameter search, which are well-established. The authors defended their approach by emphasizing the novel application of LDS for hyperparameter optimization in time-series forecasting. However, this clarification did not fully resolve the reviewer’s concerns, as the combination of LDS and LoRA still lacked groundbreaking novelty in the broader context of hyperparameter optimization.

2. Comparison with Other Tuning Methods: Several reviewers requested a direct comparison with established methods such as grid search or Bayesian optimization to better assess the effectiveness of LDS. The authors acknowledged this oversight and included additional results comparing their method with grid search and random search. However, the experimental evidence provided did not convincingly demonstrate that LDS outperforms these traditional methods, leaving the reviewer’s concerns largely unresolved.

3. Presentation and Clarity: Multiple reviewers highlighted issues with the paper's presentation, including large figures, inconsistent table formatting, and overly complex visuals that hindered the clarity of the results. In response, the authors made some adjustments to the figures and tables, though the changes did not fully address the formatting inconsistencies. The paper still lacked the professional polish expected from high-tier conference submissions.

4. Impact of LDS: A reviewer questioned whether LDS truly provided significant improvements over simpler methods, such as grid search. The authors argued that LDS improved search efficiency and performance, but the results were not compelling enough to clearly demonstrate this advantage. Despite the rebuttal, the evidence provided did not sufficiently convince the reviewer that LDS offered a tangible benefit over simpler methods.

5. Model Size and LoRA Efficiency: One reviewer noted that the paper applied LoRA fine-tuning to relatively small models, which contradicts one of LoRA's main advantages—its ability to fine-tune large models efficiently. The authors did not adequately address this concern, and the application to small models remained a limitation that was not fully justified.

6. Hyperparameter Sensitivity: A reviewer pointed out that the paper did not explore how the LDS search space’s sensitivity to hyperparameters (such as rank) affected performance across datasets. The authors briefly acknowledged this limitation but did not provide a thorough analysis or experimental evidence to address it.

---

### Decision · Program_Chairs · 2025-01-22

Reject